# User Perception of New E-Health Challenges: Implications for the Care Process

**DOI:** 10.3390/ijerph19073875

**Published:** 2022-03-24

**Authors:** María Esther González-Revuelta, Nuria Novas, Jose Antonio Gázquez, Manuel Ángel Rodríguez-Maresca, Juan Manuel García-Torrecillas

**Affiliations:** 1Grupo Investigación TIC019 Electrónica, Comunicaciones y Telemedicina (04120) Servicio Informática y Sistemas de Información, Equipo Provincial TIC, Hospital Universitario Torrecárdenas, 04009 Almería, Spain; esther.gonzalez.sspa@juntadeandalucia.es; 2Grupo Investigación TIC019 Electrónica, Comunicaciones y Telemedicina, Universidad de Almería, 04120 Almería, Spain; nnovas@ual.es; 3Unidad de Gestión Clínica Laboratorios, Hospital Universitario Torrecárdenas, 04009 Almería, Spain; manuel.rodriguez.maresca.sspa@juntadeandalucia.es; 4Unidad de Investigación Biomedica, Hospital Universitario Torrecárdenas, 04009 Almería, Spain; juanm.garcia.torrecillas.sspa@juntadeandalucia.es; 5CIBER de Epidemiología y Salud Pública (CIBERESP), 28029 Madrid, Spain; 6Instituto de Investigación Biomédica Ibs. Granada, 18012 Granada, Spain

**Keywords:** e-health, digital health, digital communication doctor-patient, e-patient

## Abstract

Establishing new models of health care and new forms of professional health-patient communication are lines of development in the field of health care. The onset of the COVID-19 pandemic has accelerated the evolution of information systems and communication platforms to guarantee continuity of care and compliance with social distancing measures. Our objective in this article was, firstly, to know the expectations of patients treated in the healthcare processes “cervical cancer” and “pregnancy, childbirth and puerperium” regarding online access to their clinical history and follow-up in the care process. Secondly, we analyzed times involved in the cervical cancer process to find points of improvement in waiting times when digital tools were used for communication with the patient. A descriptive cross-sectional study was carried out on 120 women included in any of the aforementioned processes using a hetero-administered questionnaire. The analysis of times was carried out using the Business Intelligence tool Biwer Analytics^®^. Patients showed interest in knowing their results before the appointment with the doctor and would avoid appointments with their doctor if the right conditions were met. Most recognized that this action would relieve their restlessness and anxiety. They were highly interested in receiving recommendations to improve their health status. It was estimated that there was room for improvement in the times involved in the care process, which could be shortened by 34.48 days if communication of results were through digital information access technologies. This would favor the optimization of time, resources and user perception.

## 1. Introduction

The current moment is characterized by continuous digital evolution. Our habits, way of working, communication and personal relationships in general have changed towards a new digital scenario thanks to the new technological ecosystems that have been generated.

The incorporation of Information and Communication Technologies (ICT) in any field is a process that represents an advance in work methodologies, response times, results and, in general, a development of the set of elements that make it possible [1]. The same is true in the field of health care. The use of ICT, as long as it is carried out properly, with criteria of quality and efficiency, will mean an improvement for the user, the professional and the health system as a whole. Documented analyses of the main barriers to and enablers of the implementation of e-Health services are detailed in [2], which lists ten fundamental barriers and six main enablers.

New technology based on their use through computer tools and mobile devices are causing a change in the doctor-patient relationship and actions by the healthcare center-citizen binomial. The ways of communicating and interacting and the need for the multidirectional transfer of information must be incorporated into the use of new tools. This implies addressing new challenges and dilemmas on the part of the organization, which entails the incorporation of new methodologies in their work models [3].

Some of these emerging technologies are the use of blockchain, artificial intelligence (AI), quantum computing, the medical internet of things (MIoT) and Big Data. Each of these technologies provides us with an opportunity to advance and improve in all areas of health and in digital medical care in particular. Many cases of use of these technologies can be found in the bibliography; we comment on some relevant ones.

MIoT facilitates the online monitoring and control of patient clinical data, such as medical images, heart rate, blood oxygen level, medication dose and history of health conditions, among others [4,5].

AI facilitates clinical decision support, patient management automating service provision, patient monitoring and health interventions [6].

Blockchain is a technology that ensures the security of the data collected and helps maintain your privacy. Various technologies such as industry 4.0, bio-sensors, 3D scanning, and multi-agent systems have been used in telemedicine for diabetic patients in the fight against the COVID-19 pandemic. Others include control of infectious diseases and prevention of the spread of the pandemic through contact tracing and movement of people. In March 2020. The World Health Organization (WHO) launched MiPasa, which is a platform based on blockchain technology that facilitates the totally private exchange of information between individuals, state authorities and health institutions [4,7].

Combining the development of these technologies with the use of appropriate tools will facilitate the exchange and interaction of information in real time.

Motivated on the one hand by the demands of users who handle technology more and more in their day-to-day life and given that the tools are increasingly accessible to anyone, organizations and companies themselves understand that they must respond to this demand and need; in relation to this, they are immersed in a process of analysis and evaluation of new developments to incorporate and promote the use of ICT in their operation [8]. The forms and means of contact between patient and healthcare professionals will increasingly be oriented towards the use of mobile devices. This, together with the increase in developer applications (APPs) [9], make it necessary to advance on this path and offer patients with resources and tools that allow them to have more direct and interactive communication, developing new relationship models, promoting autonomy of the patient and facilitating knowledge and interaction with their care process through ICT. In this context, one of the critical points in patient safety is the use of mobile devices to monitor their health status [10].

Communication between health professionals and patients plays a decisive role in the development and evolution of any healthcare process [11]. Adequate communication in a timely manner will provide a decrease in patient anxiety, a degree of improvement in adherence to treatment and interpretation of results, a minimization of additional diagnostic tests, a reduction in the number of claims, optimization of time and resources, improvement in the management of visits and, in general, an improvement in the patient’s perceived quality of the care process. The methodology used to carry out this communication is therefore also of vital importance throughout the process. Adapting it to new technologies is already a stated objective and an object of analysis and discussion [12].

New technologies tend to be assimilated with increased usefulness and efficiency, but some unknowns should be answered beforehand: do users have the same perception? What do users really think of the use of ICT to know the status of their care process? Do they use digital media to access their health histories? To what extent do they agree to replace current media with newer ones? It is considered important and convenient to analyze the proposals that are raised with methodological rigor and a critical spirit by assessing the opinion of users, which are one of the points of analysis and development in this work.

Through a survey, we evaluated the opinion of users regarding the use of digital tools to access their clinical information, specifically access to test results through tools such as APPs, access to Web Health portals, SMS, instant messaging or email. The analysis carried out for the process of cervical cancer can be extrapolated to many other processes in the field of health care.

We also evaluated the interest users presented in relation to several specific use cases; that is, at what point in their care process do they consider access to their information may be of greater interest to them.

On 11 March 2020, the outbreak of the disease caused by the corona-virus SARS-CoV-2 (COVID-19), was evaluated as a global pandemic by the World Health Organization (WHO). As a consequence of the activation of the pandemic, a series of provisions and containment measures were put in place in the face of the emerging health crisis (Order SND/234/14 March 2020) [13]. This has accelerated the implementation of measures and new forms of health care for patients that guarantee compliance with the measures imposed by the WHO. The development of patient communication platforms, information systems and support for decision-making has been promoted, in addition to new digital tools that facilitate communication and accessibility with the aim of preserving continuity in care, while containing the spread of the pandemic [14]. The field of informatics and biomedicine is playing a very prominent role in the fight against COVID-19 [15].

This work represents a feasibility and opportunity study. Therefore, the final objective was to know the opinion and expectations of users in relation to the use of information and communication technologies for the follow-up of their care process and the knowledge of their clinical health history. Having recognized the need to implement and promote digital platforms for the exchange of information in real time and whose progress has been accelerated by the pandemic, we focus on a specific practical use case, the cervical cancer care process, and calculate times involved in the care process, among them, the time of communication of the test results to the patient. We determined the bounding of the response times involved in the study, how they have been affected by the COVID-19 pandemic and how we could shorten the duration of a process and eliminate waiting times by making use of digital media to which we have referred.

## 2. Materials and Methods

The scope of this study focused on two care processes in specific areas of knowledge and hospital care, known as integrated care processes (ICPs). In this case, the ICPs “cervical cancer” and “pregnancy, childbirth and puerperium” were addressed [16,17]. A descriptive and cross-sectional study of the patients who attended the gynecology consultation and another group of patients who attended the laboratory consultation (extraction room) was carried out. Patients in the laboratory group came for gestational diabetes screening, as part of the universal screening protocol included in the PAI pregnancy, childbirth and puerperium for those patients with previously altered blood glucose levels. Patients in the gynecology group came for gynecological examination after clinical or diagnostic suspicion and undergo a molecular test for the detection of human papillomavirus (HPV).

The selection of the participants in the study was carried out randomly among all users who made use of any of the consultations during the period of patient inclusion from January to June 2018.

The sample size was estimated from a target population of 540 patients. To achieve a precision of 5% in the estimation of a proportion using a normal asymptotic two-sided 95%, confidence interval with correction for finite populations, assuming that the expected proportion of acceptance of ICT is 75% and that the total size of the population is 540, it would be necessary to include 120 patients in the study.

Subsequently, an analysis of the times involved during the cervical cancer care process was carried out. For this, two periods of analysis were chosen, of similar patient populations: a pre-pandemic period, from June to October 2019, with 227 patients and a period immersed in a pandemic, from May to July 2021, with 239 patients. To carry out the surveys, the identity of the patient was not revealed at any time, so it was not necessary to collect consent.

### 2.1. Methods

To respond to the objective of this work, a questionnaire was administered to 120 women randomly selected from among the consultants of both ICPs (pregnancy, childbirth and puerperium, as well as cervical cancer). The surveys were conducted by qualified persons.

A preliminary pilot test was carried out to homogenize and agree on the way in which the questionnaires were administered and thus reduce variability. There was no refusal to participate in the face of requests to answer the questionnaire (Table 1).

A Likert-type scale with five categories was used, including “Totally disagree/I am not interested in anything”; “Partially disagree/Not very interested”; “Indifferent (neither agree nor disagree)”; ”I partially agree/I am interested in something”; “Totally agree/I’m very interested.” These answers, evaluated as categories 1 to 5 respectively, were considered as ordinal for the statistical analysis [18].

The questionnaire was anonymous and distributed randomly (systematic random sampling with a choice of 1 out of 5 patients) among those who attended consultations until the calculated sample size was achieved.

The development of the survey was based on the TAM methodology (Technology Acceptance Model) [19]. It establishes the degree of acceptance by society before the introduction of new technologies and how the population reacts to changes. Elements such as perceived usefulness, perceived ease, attitude, and intention to use are valued. The TAM model is applicable to the object of development and analysis. Regarding the perceived usefulness, the highlighted question is: do users think that with the use of new technologies to monitor their process or their clinical history in general they will have improvements?

The issues were grouped into various dimensions or areas of interest. On the one hand, sociodemographic variables (sex and age), degree of interest in the use of technology to know results, in which cases would it be more interesting to use telematic means, which means they prefer and in which of them they would have more confidence to obtain information on their clinical history in general and their care process (email, mobile applications, web applications, SMS and others). On the other hand, it was also questioned whether the patient would dispense with his appointment with the doctor if once the result of any of the tests of the process was known and the consultation with the doctor was not strictly necessary.

In addition, the integrated care process of ‘cervical cancer’ was analyzed using a Business Intelligence (BI) tool. This tool allowed us to disaggregate the information at a high level of detail. We carried out a real analysis of the times involved in patient care throughout the process (entry-exit), applied to two periods, pre-pandemic and during the pandemic. The times analyzed were evaluated from the time the patient was summoned and goes to the gynecology consultation (1st appointment); a molecular test was required for the detection of HPV in a sample of cervical exudate, which was sent to the laboratory responsible for its processing. When the laboratory finished and validated the study, the patient was scheduled again for the gynecology consultation (2nd appointment) to communicate the results of the laboratory report and continue the care process based on the algorithms defined in the ICP.

### 2.2. Statistical Analysis

Statistical analysis was performed using SPSS statistical software. In the descriptive analysis, the qualitative variables are expressed as frequencies and percentages. Quantitative variables are expressed as meanS, standard deviationS (SD) or medianS accompanied by the interquartile range (IQR).

For the comparison between qualitative variables, Fisher’s exact test or Pearson’s chi-square test was used. For the comparison of the ordinal variables on the Likert scale of the questionnaire, the Mann-Whitney U test was used. All tests were performed with 95% confidence intervals.

Those variables where statistical significance was found are indicated with a (*), and those others with indications of statistical significance are indicated with a (&).

On the other hand, to carry out an analysis of the real times involved in the cervical cancer process, we defined several control points that make up part of the process, as can be seen in Figure 1.

The times involved in the process, as detailed in Figure 1, are conceptualized as follows:

MTR: mean time of HPV test request—the time in days that elapses from when the patient has her appointment in the gynecology consultation until the request for the HPV test is registered in the laboratory information system.

MTV: mean validation time—the time in days that elapses from when the request for the HPV test is registered in the laboratory information system until the test is validated by the laboratory, which means that the result is available.

MT2C: mean time of second consultation—the time in days that elapses from when the test result is available once validated until the patient has her appointment at the hospital to inform her of the result and continue the care process based on the result.

These times were calculated for patients with appointments for consultations in two periods to analyze a period prior to the pandemic (June to October 2019), with 227 cases of analysis, and a subsequent period once the pandemic was activated (May to July 2021), with 239 cases. The calculation periods included were selected as they were considered to be representative and the data obtained was extrapolatable to other periods, the results of which have been obtained in similar proportions.

The analysis was performed using the Biwer Analytics^®^ BI tool with data from different sources, the Laboratory Information System (LIS), the appointment management information system, as well as the hospital information system.

Anonymized records were used to extract the data from the medical records, eliminating the identifying data. The guidelines provided by the ethics and research committee of the province of Almería (Spain) were followed.

## 3. Results

Of the total number of patients surveyed, 53.3% (*n* = 64) came from the gynecology consultations compared to 46.6% (*n* = 56) coming from the general laboratory. Age groups related to fertility and diagnosis of cervical cancer were considered. From the point of view of age distribution, the most prevalent group of patients who came from the laboratory consultation was the youngest group (≤35 years old), while the most prevalent for those from the gynecology consultations was the 36–65 year-old age group. Regarding the age distribution, statistically significant differences were found between the origin of consultations and the age groups analyzed (*p* < 0.001) (Figure 2).

The questions whose average score is greater than 4 (I’m partially agree/I am interested in something), considering that the assessment is made on a Likert-type scale from 1 to 5 (Totally disagree/I am not interested in anything to Totally agree/I’m very interested), included mainly five: P4, P6.3, P7, P9 and P10.

To the question: Would you be interested in knowing your results before your doctor’s appointment? (P4) It was observed that a majority of patients were interested in knowing the results before going to the face-to-face consultation with their doctor, presenting an overall mean of 4.08 (Table 2).

For the question: What other information related to the status of your analytics or your test would you like to know? related to “Results are now available” (P6.3), interviewees were interested in knowing the results before their appointment with the doctor with a score of 4.37 (Table 2).

For the question: Do you think that advance knowledge of the status of your analysis process and its results would alleviate your anxiety while waiting? (P7), the response reflects that the interviewees were interested in information related to the availability of the results of their tests, presenting an overall mean score of 4.08 (Table 2).

For the question: Would you like to avoid unnecessary appointments with your doctor, when your results were normal and you did not need any medical intervention, as long as the result was accompanied by an explanatory report? (P9), the response reflects that the interviewees would be interested in receiving the information avoiding unnecessary appointments with a score of 4.16 (Table 2).

For the question: Would you be interested in receiving information accompanied by laboratory reports on lifestyle habits, hygiene, care, etc., that would help you better control your disease? (P10), the response had a score of 4.61, indicating that the patients surveyed were interested in receiving recommendations to improve their state of health through healthy lifestyle habits (Table 2).

In a complementary way, Table 3 presents a summary of the percentages of respondents grouped for each option.

When the results were analyzed in more detail and those of one query were compared with the other (Table 4), interesting results were obtained, among which we highlight the following:

Regarding the first variable P1 (level of knowledge of the patient in relation to whom and how they will be informed of the results of their test), the score was higher in the patients who went to have the test performed at the laboratory than in the patients who went to the gynecology consultation (3 vs. 4), *p* ≤ 0.05.

On the other hand, statistically significant differences were detected in the variables P3, P4 and P56 (*p* ≤ 0.05). In these items, the same median value was found for both strata; however, their distribution behaved differently (Figure 3).

For each visit (gynecology and laboratory), we analyzed whether there were differences between age groups (≤36 years vs. >36 years) for each of the items (Table 5). For the gynecology consultation, a statistical significance was observed for the variable P5.2. In addition, items P1 and P5.1 showed signs of statistical significance with higher scores in patients ≤35 years. Finally, for the laboratory consultation, statistically significant differences were found with respect to age for the item (1 vs. 0, *p* ≤ 0.05). Item P8 presented higher values in those ≤36 years old (5 vs. 1, *p* ≤ 0.05).

After calculating the different periods analyzed in the care process, we obtained homogeneous and significant data (Table 6). The MTR is 15 days, the MTV is 9 days and the MT2C is 34.48 days.

## 4. Discussion

This study highlights the interest that patients present a priori in relation to the use of new technologies to monitor their care process, as well as to the amount of information about their next contact with the professional. In short, it indirectly exposes patients’ involvement in their care process, facilitating the doctor-patient relationship through new means of communication where new technologies prevail with computer tools for on-line access to their care data and recommendations for health habits and healthy living directly related to their illness or treatment [20].

In the analysis, two types of tests are differentiated: one performed in the laboratory consultation aimed at patients whose test is for the detection of gestational diabetes as part of the universal screening protocol within PAI pregnancy, childbirth and puerperium, and who have had previously altered blood glucose levels. The other case is for patients with some risk factor or clinical suspicion who came from a gynecology consultation (PAI cervical cancer).

The operative process (clinical-care) in both cases begins after the person’s contact with the health system through the different possible entrances, Primary Care (PC) or Hospital Care (HC). Considering the previous system, the care that professionals offer from different areas of action in PC and HC could be further supplemented by the continuity of patient/family care [21].

To adequately evaluate the results obtained, it is necessary to note that cervical cancer is the tenth most frequent cancer among Spanish women, and the second among 15–44-year-olds [22]. A woman diagnosed with cervical cancer does not abandon the care chain, as she requires regular follow-up. Patients who show signs of clinical and/or diagnostic suspicion after a gynecological examination carried out in consultations will undergo cytology. When faced with a pathological cytology, a series of actions are triggered, including an HPV test, aimed at a diagnosis and treatment that should be carried out within a reasonable period. This period on certain occasions is long, with the aggravating circumstance that the patient is unaware of the results. Integrating the use of new technologies into the process could be key and help in initiatives such as screening programs and other routine processes [23]. Consultations whose information can be offered online to the patient could be reduced in favor of those that present cytological alterations and those that should be streamlined.

We start from the fact that this group of patients was specifically taken as the object of study, considering that the development of the phases of the process admitted the possibility of incorporating the complementary use of ICT and new digital media.

For this type of patient, the times involved in part of their care process were calculated. The results reveal the existence of a significant time gap that could be avoided or minimized; this gap runs from when the test that is requested from the patient (HPV) is performed and validated by the laboratory until they have the next appointment at the hospital for the gynecology consultation (MT2C), when the result of the test is communicated. and the process continues. The average time that elapses is 34.48 days. In cases in which there is no cytological alteration or complication, the result could be made available to the patient from the moment the test is validated by the laboratory, in this way, the use of digital media and ICT would contribute to reducing the patient’s waiting time, reducing the number of second visits, and avoiding face-to-face visits as much as is convenient, something that is recommended by the authorities in times of crisis, as is the current case in favor of containment of the spread of the pandemic. These freed spaces could be occupied by patients on the waiting list whose appointments were delayed due to the high volume of patients waiting to be seen.

In the case of the ICP for pregnancy, childbirth, and the puerperium, there is an extensive program of activities to follow up in an integrated manner between PC and AH, establishing a series of visits, appointments, and recommendations throughout the process that encourage the participation of women in its development. It is therefore feasible for technology to be used throughout to support and improve this process [24,25].

Providing the factors, information, and motivation for professionals to promote new means of communication and transmission of results are a determining factor for progress in this line [26].

In the present work, the attitude and predisposition towards the use of ICTs is considered, assessing whether patients will be willing to incorporate the change in their form of contact with the health specialist and whether they will have a proactive attitude [9,27]. Results presented by other authors who have conducted patient surveys show that more than 60% of the respondents were interested in the possibility of accessing their Electronic Health Records in general and even 50% of them were willing to pay for it [28].

The results and previous studies support a clear interest on the part of the population in receiving recommendations, for example, to improve their health status through healthy lifestyle habits. A total of 54.2% of the general population seeks information on food, nutrition, or healthy lifestyles [12]. The score of the patients analyzed was one of the highest, recognizing the degree of interest with an average score of more than 4.6. Therefore, there is a clear congruence between the population-based findings and those from this study sample.

In this area, the development and promotion of the use of mobile devices, such as wearables, would facilitate both the reception of this information and the transmission of constants to the health professional, which would allow better diagnoses to be made and the recommendations to be adapted and focused on the case of each patient [29,30].

Even though almost 85% of the population in the 25–34 year-old age group is aware of the existence of devices that can measure and provide data on their health status, they are only used by one in four women, especially those with chronic pathology.

It should be noted that the use of mobile applications (APPs) by patients to monitor their health is only 4.3%, with a little more notoriety being used for monitoring physical activity, healthy living, and well-being of personnel, with 8.6%. This low interest is reflected in the analyzed surveys, where the score in relation to the preference in knowing the results of their care process or of their test through an APP has one of the lowest values (1.44 mean). The health professional does not yet have a high level of confidence in this type of application. The percentage of health professionals who have recommended websites is only 6.4% and 7.5% in the case of APPs [1]. From the foregoing, we can infer that although users are interested in using the new communication channels, they are somewhat reluctant to use them regularly, probably waiting for their doctor to be the one to transmit and promote it.

Patients in general have difficulties when viewing their clinical information through portals, APPs and the use of new computer tools in general, mainly due to the lack of ease of navigation, interpretation of results, ignorance of the parameters, if they are included in a range of normality or not, next steps, etc. [31]. The lack of validity, reliability, rigour, accreditation, usefulness, relevance, credibility, accuracy, and ignorance of the tools themselves, are some of the limitations for the expansion in the use of these technologies both for patients and professionals.

Just over half of the people who have used them confirm that their use has avoided unnecessary visits to the hospital or health center, which leads to cost savings for the Health System and an improvement in the management of appointments face-to-face. In addition, 55.2% agree that these technologies have helped them improve or prevent health problems [32].

The results show that there is an interest in knowing the data as soon as possible (knowing the results in advance would alleviate the anxiety of the patients with a score of 4.08 in item P7), even without a face-to-face appointment; however, it cannot be inferred which would be the most appropriate method, since when asking about the different technological means (APPs, WhatsApp instant messaging, SMS, Web, etc.) there is no determining score (all oscillate around an average of 2.5 out of 5).

Regarding the degree of interest, it is also observed that there is a difference between the patients in the laboratory consultation and those who present some risk factor or suspected pathology (who attended the gynecology consultation). In both cases, the value of the median is 5 (Totally agree/I’m very interested), indicating that they are interested in knowing the results before the appointment; however, those coming from a laboratory consultation have more knowledge of the times of their process with a value of the median of 4 (I partially agree/I am interested in something) points for the knowledge of how and who will inform him and 5 (Totally agree/I’m very interested) for the knowledge of the date of his next appointment. Possibly in these cases the information will be given by her PC doctor and therefore the patient knows better who and how she will be informed following the traditional procedure; in addition, they show greater interest in the use of other technological means that speed up and advance information. Those considered unhealthy or immersed in a more complex process prefer direct information from their doctor [33].

Some of the analyzed variables are conditioned on age. When stratifying by age, specifically, those over 36 years of age who attend the gynecology consultation usually have less information about who and how they are going to be informed. In relation to the above, all patients show a high interest in knowing the results before the appointment, although women who attended the gynecology consultation and especially those over 36 years of age showed median values of 0 and 1, respectively. They have little interest in knowing the results via email, SMS, instant messaging, mobile applications, or other digital means. This could be since these patients are usually immersed in more complex processes, with a risk factor, and prefer to receive a direct explanation from their doctor, not only because of the difficulty in understanding the results, but also because of the anxiety and fear that they carry with them about receiving a diagnosis that could be positive or negative. However, laboratory patients that are asymptomatic with no risk factor have a banal process in the highest percentage of cases.

Regarding the means that laboratory patients prefer to know the results, with a median of 4.5 through Web pages for those over 36 years of age and with a median of 5 through instant messaging. This is currently the most widespread means for this type of case.

The limitations of this study are fundamentally found in the fact that the population under study includes only women, and gender can be considered a determining factor. The specialties and processes chosen made it possible to compare the involvement of the patient in a banal process versus a more complex process. However, contrasting the results of this study with those of others, it can be deduced that they can be extrapolated to other processes with similar characteristics, although there are other factors that can also be considered decisive in the results, such as sociodemographic indicators, pathology, chronicity, sex, cultural level, socio-professional situation and others [34]. It remains therefore as future work to propose the expansion of the analysis under study in the aforementioned context.

The use of telemedicine, telehealth and, in general, digital means for accessing and monitoring healthcare processes, as well as clinical history in general, has been incentivized by the pandemic. Simply, patient care systems have the potential to take advantage of information technologies. This crisis has been an opportunity to promote the use of information technologies in the health field [35].

The need to adapt information systems that support citizen health in general and digital health histories has been highlighted, although their acceleration has caused situations of deregulation and has made visible the deficiencies and limitations in guaranteeing the safety, privacy and general quality of care [10,36,37].

The level of acceptance, continuity and progress of the new media and services promoted remains to be verified and whether the progress produced will be a solid base for continuing to advance in this line at a sufficient pace offers great potential, but at the same time also has great limitations that will have to be addressed. Given the overload of care demand that the health system is experiencing, it is very important to distribute the workload among professionals efficiently and safely. The analysis that we have carried out in this work and the current situation provides objective information on the need to continue advancing in the development of new forms of health care and promotion using digital media and, in general, the promotion of information technologies and communication for eHealth [38].

Finally, we must point out some limitations to the above. These include barriers responsible for inhibiting the development and application of information technologies in health care. Some of the variables related to these barriers are the efficiency of doctors and care professionals, patient preferences for receiving face-to-face care, an inadequate legal framework, a deficiency of explicit standards, lack of adequate resources, inadequate telecommunications technology and infrastructure, interoperability, implementation problems, socio-administrative problems, violations of privacy and confidentiality, and insufficient research in this area [39].

## 5. Conclusions

The development of care processes allows for a comprehensive approach to health problems. The inclusion of new technologies and digital progress incorporates key improvements in the development of the processes.

eHealth applications offer the patient a possibility for improving the control and monitoring of their health status, streamlining circuits, bringing them closer to the treatment of their process and involving them through new tools. Knowing the opinion of users is essential for their evolution. We focused our study on the analysis of the opinion of patients involved in two high-impact care processes.

Moving towards a doctor-patient digital ecosystem, taking advantage of the opportunities that technology offers, and prioritizing the patient’s quality of life would allow optimization of time and resources for both the professional and the patient in situations that the health professional deems appropriate. Using a Business Intelligence tool, we analyzed the duration of a process and identified time intervals that can be reduced. For those situations in which the absence of disease must be communicated without the need to provide other types of complex clinical information, unnecessary consultations could be avoided and users could receive the information accurately and with the immediacy of digital environments. This would result in a reduction in anxiety and concern of patients, shortening unnecessary waiting times. It would also help reduce the number of face-to-face consultations, improving the availability of resources in the field of care.

All these means and actions are complementary ways and tools to support the healthcare work of the professional, while always safeguarding the fundamental rights of information, intimacy and confidentiality of the patient.

The evolution of the e-informed patient and the “health professional-patient” relationship represents one of the future challenges in health care.

With the onset of the pandemic, in many cases it has been the health professionals themselves who have seen the potential offered by the use of technology for care in circumstances such as those experienced, and who have facilitated the expansion and use of telematic health systems. However, the extraordinary expansion of the implementation of telehealth and its use cases highlights the need to carry out an exhaustive analysis of the real effects produced in patients and health models.

The use of information technologies and analysis through Business Intelligence tools as detailed in this article help us obtain results that guide us to take measures and implement actions that address the improvement of health care, minimize response times, facilitate accessibility and optimize available resources.

## Figures and Tables

**Figure 1 ijerph-19-03875-f001:**
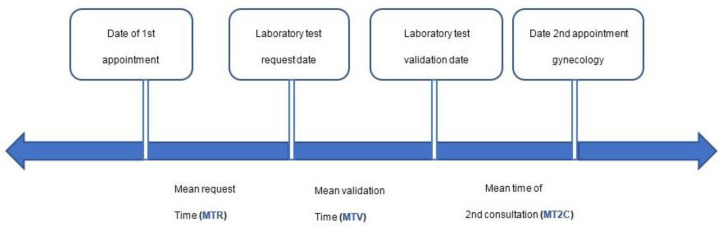
Times involved in part of the cervix cancer care process.

**Figure 2 ijerph-19-03875-f002:**
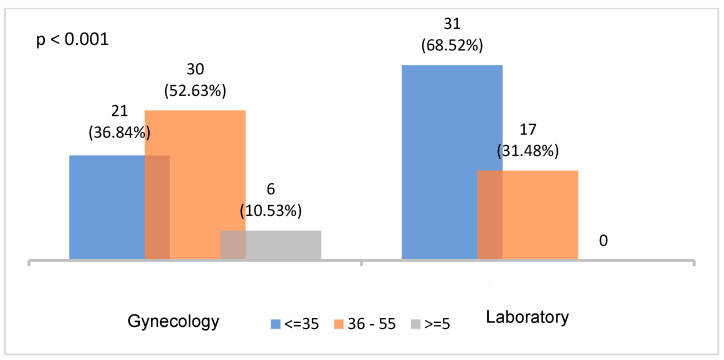
Age groups according to type of consultations (*p* < 0.001).

**Figure 3 ijerph-19-03875-f003:**
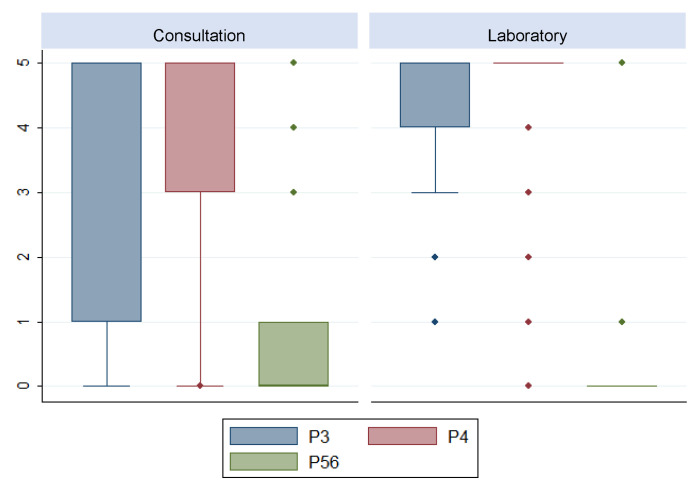
Box-Plot of the items 3, 4 and 56.

**Table 1 ijerph-19-03875-t001:** Patient Survey.

	Tell Us Your Age before You Start		Year				
P1	1	Do you know how or who will inform you of your test results?	1	2	3	4	5
P2	2	Do you know how long your results will take?	1	2	3	4	5
P3	3	Do you already have the date of the next appointment with your doctor to inform him of the results?	1	2	3	4	5
P4	4	Would you be interested in knowing your results before your doctor’s appointment?	1	2	3	4	5
P5	5	By what means would you like to be able to access your lab results?					
P5.1	Email (Gmail, etc.)		1	2	3	4	5
P5.2	SMS		1	2	3	4	5
P5.3	Whatsapp type instant messaging		1	2	3	4	5
P5.4	Access to a secure WEB page		1	2	3	4	5
P5.5	Through an APP		1	2	3	4	5
P5.6	Others Tell us how		1	2	3	4	5
P6	6	What other information related to the status of your analytics or your test would you like to know?					
P6.1	The sample has arrived correctly at the laboratory		1	2	3	4	5
P6.2	The analysis is being processed by the laboratory equipment		1	2	3	4	5
P6.3	Results are now available		1	2	3	4	5
P6.4	An error has occurred in the process and you must contact the laboratory		1	2	3	4	5
P7	7	Do you think that advance knowledge of the status of your analysis process and its results would alleviate your anxiety while waiting?	1	2	3	4	5
P8	8	Would you like to avoid unnecessary appointments with your doctor, when your results are normal and you do not need any medical intervention?	1	2	3	4	5
P9	9	Would you like to avoid unnecessary appointments with your doctor, when your results were normal and you did not need any medical intervention, as long as the result was accompanied by an explanatory report?	1	2	3	4	5
P10	10	Would you be interested in receiving information accompanied by laboratory reports on lifestyle habits, hygiene, care, etc., that would help you better control your disease?	1	2	3	4	5
P11		Want to help us start this project?	YES		NO		

**Table 2 ijerph-19-03875-t002:** Description of age and survey items.

	Total	Item Description
Consultation Gynecologal *n* (%)	64 (53.33)	Origin of the respondent
Laboratory *n* (%)	56 (46.67)	
Three age ranges		
≤35 *n* (%)	58 (48.3)	
36–55 *n* (%)	47 (39.2)	
>55 *n* (%)	6 (5)	
Unknown *n* (%)	9 (7.5)	
Ítems	Average (Standard deviation)	
P1	3.30 (1.67)	Know how/who will inform you
P2	3.58 (1.54)	Know the waiting time for results
P3	3.86 (1.68)	Know the appointment date for results
P4	4.08 (1.59)	You are interested in knowing results before the appointment
P5.1	2.94 (2.24)	You prefer to know the result by email
P5.2	1.90 (2.12)	Preference for knowing the result via SMS
P5.3	2.21 (2.21)	Preference for knowing the result via instant messaging (WhatsApp, others)
P5.4	2.57 (2.56)	Preference for knowing the result through the website
P5.5	1.44 (1.89)	Preference for knowing the result through the app
P5.6	0.66 (1.56)	Preference for knowing the result by other means
P6.1	2.93 (2.04)	Interest in knowing if the sample is in the laboratory
P6.2	2.35 (1.99)	Interest in knowing if the sample is processed
P6.3	4.37 (1.41)	Interest in knowing if the results are available
P7	4.08 (1.26)	Anticipating knowledge of sample status and results alleviates anxiety
P9	4.16 (1.39)	If the results are normal and accompany a report, do you think you would avoid a medical appointment?
P10	4.61 (0.94)	Interest in receiving information on healthy lifestyle habits related to your case

**Table 3 ijerph-19-03875-t003:** Frequency of questionnaire items.

Variables	Response	Gynecology		Laboratory		Variables	Gynecology		Laboratory	
		Count	%	Count	%		Count	%	Count	%
P1	1	21	32.8	10	18.2	P6	25	12.0	12	6.8
	2	5	7.8	4	7.3		8	3.8	14	7.9
	3	11	17.2	4	7.3		18	8.7	20	11.3
	4	7	10.9	11	20.0		23	11.1	18	10.2
	5	20	31.3	26	47.3		134	64.4	113	63.8
P2	1	16	25.0	5	8.9	P7	6	9.4	4	7.1
	2	6	9.4	6	10.7		3	4.7	0	0.0
	3	6	9.4	8	14.3		10	15.6	11	19.6
	4	9	14.1	13	23.2		5	7.8	15	26.8
	5	27	42.2	24	42.9		40	62.5	26	46.4
P3	1	17	27.0	7	12.5	P8	13	20.6	6	11.1
	2	3	4.8	3	5.4		5	7.9	3	5.6
	3	4	6.3	1	1.8		9	14.3	2	3.7
	4	3	4.8	5	8.9		0	0.0	9	16.7
	5	36	57.1	40	71.4		36	57.1	34	63.0
P4	1	8	13.3	3	5.6	P9	9	14.1	4	7.1
	2	2	3.3	1	1.9		3	4.7	3	5.4
	3	7	11.7	4	7.4		7	10.9	5	8.9
	4	3	5.0	2	3.7		4	6.3	3	5.4
	5	40	66.7	44	81.5		41	64.1	41	73.2
P5	1	57	28.2	32	17.1	P10	3	4.7	1	1.8
	2	11	5.4	17	9.1		1	1.6	1	1.8
	3	13	6.4	24	12.8		4	6.3	4	7.1
	4	10	5.0	16	8.6		4	6.3	5	8.9
	5	111	55.0	98	52.4		52	81.3	45	80.4

**Table 4 ijerph-19-03875-t004:** Scores by Origin [Medians (RIQ) and *p*-value]. * Statistically significant differences.

Variables	Gynecology	Laboratory	*p*
P1	3 (4)	4 (3)	≤0.05 ^(^*^)^
P2	4 (3.5)	4 (2)	0.278
P3	5 (4)	5 (1)	≤0.05 ^(^*^)^
P4	5 (2)	5 (0)	≤0.05 ^(^*^)^
P5.1	2 (5)	5 (5)	0.207
P5.2	1 (5)	1 (4.5)	0.679
P5.3	1 (5)	2 (5)	0.435
P5.4	1 (5)	3.5 (5)	0.141
P5.5	0 (2.5)	1 (3)	0.550
P5.6	0 (1)	0 (0)	≤0.05 ^(^*^)^
P6	4.5 (1)	5 (0)	0.480
P6.1	4 (4)	3 (5)	0.645
P6.2	2 (5)	2 (4)	0.561
P6.3	5 (0)	5 (0)	0.771
P6.4	5 (4)	5 (3)	0.940
P7	5 (2)	4 (2)	0.355
P8	5 (3)	5 (1)	0.326
P9	5 (2)	5 (1)	0.246
P10	5 (0)	5 (0)	0.982

**Table 5 ijerph-19-03875-t005:** Scores according to origin and age [Medians (RIQ) and *p*-value]. * Statistically significant differences and those others with indications of statistical significance are indicated with a (&).

	Ginecology			Laboratory		
Variables	Age ≤ 36	Age > 36	*p*	Age ≤ 36	Age > 36	*p*
P1	3 (3)	2 (3)	0.083 ^(&)^	4 (3)	4.5 (2)	0.786
P2	5 (4)	4 (3)	0.549	4 (2)	5 (1)	0.147
P3	5 (4)	5 (3)	0.530	5 (1)	5 (1)	0.989
P4	5 (2)	5 (4)	0.214	5 (0)	5 (1)	0.561
P5.1	5 (4)	1 (5)	0.065 ^(&)^	5 (4.5)	5 (1)	0.622
P5.2	1 (4)	0 (1)	≤0.05 ^(^*^)^	2 (4.5)	1 (5)	0.963
P5.3	1 (5)	1 (5)	0.413	1.5 (5)	5 (4)	0.165
P5.4	3 (5)	0.5 (5)	0.180	1 (3)	4.5 (1)	0.352
P5.5	1 (4)	0 (1)	0.126	0 (0)	0 (1)	0.383
P5.6	0 (1)	0 (5)	0.485	5 (0)	0 (0)	0.369
P6	-	5 (0)	-	3.5 (4.5)	-	-
P6.1	4 (3)	3 (4)	0.550	3 (4)	2.5 (4)	0.248
P6.2	3 (4)	1.5 (5)	0.409	3 (4)	2 (4)	0.553
P6.3	5 (0)	5 (0)	0.957	5 (0)	5 (0)	0.878
P6.4	5 (1)	5 (4)	0.612	5 (3.5)	5 (3)	0.762
P7	5 (2)	5 (2)	0.940	4 (1.5)	4 (2)	0.600
P8	5 (3)	5 (4)	0.930	5 (1)	3.5 (4)	≤0.05 ^(^*^)^
P9	5 (2)	5 (2)	0.985	5 (0)	5 (2)	0.259
P10	5 (0)	5 (0)	0.853	5 (0)	5 (0)	0.401

**Table 6 ijerph-19-03875-t006:** Cases analyzed with time delay (in days) during the care process.

	Number of Cases (1st Visit Number of Patients)	Number of Patients (1st Visit Requiring) HPV	No. of Patients with 2nd Visit	MTR (Days)	MTV (Days)	MT2C (Days)
1st Períod	227	144	137	16.36	8.26	39.70
2nd Períod	239	104	94	11.84	9.80	31.50
Both periods	466	248	175	14.66	8.87	34.48

## Data Availability

The data presented in this study are available on request from the corresponding author. The data are not publicly available due to ethical restrictions.

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
