# Peer review of "User Perception of New E-Health Challenges: Implications for the Care Process"

_ijerph, 2022, doi:10.3390/ijerph19073875_

Round 1

Reviewer 1 Report

Esther Gonzalez et.al., aim to understand the opinion and expectations of the users in relation to the use of Information and Communication Technologies for the follow-up of their care process and the knowledge of the state of their Clinical Health History. The manuscript is organized well; however, some parts must be explained clearly. Following are my major concerns to this paper:

  1. Abstract need to be improved. It should be clear and concise. For example, Line 22-25 the sentences are long. The objective of the article is not clearly defined. In line 27, be specific with Business Intelligence tool.
  1. Introduction lacking information. Authors must include about the technologies that facilitates healthcare systems such as the role of blockchain, AI and MIoT. Please refer the following manuscript. A Blockchain and Artificial Intelligence-Based, Patient-Centric Healthcare System for Combating the COVID-19 Pandemic: Opportunities and Applications.
  1. As mentioned in the line 100-102, “……how they have been affected by the COVID-19 pandemic and how we could shorten the duration of a process, eliminating waiting times by making use of digital media”. My question is: Did authors really addressed this concerns? Please clarify in detail.
  2. Abbreviations missing for several words such as IAP, TIC, EPI, AGE, TMP, TMV, TM2C and much more. Authors must correct these errors.
  3. Table 1 hard to follow. What is P51 to P55? If it is subsection of P5 represent appropriately. Why there is year and ----------?
  4. Page 4, line168 “…. questionnaire were shown as mean (SD) in each of the dimensions analyzed, …..” In general SD stands for Standard deviation and authors mentioned mean.
  5. How the data's are obtained for the analysis? In Section 2, Authors claimed that data obtained from January to June 2018. Whereas in statistical analysis section, Line 193-195, authors claimed that (June to October 2019) and (May to July 2021). Recommending authors to mention clearly about all the consultation periods in the Section 2.
  6. Define the evaluation metrics.
  7. How this study helps to improve the patient care using existing technologies and give some use cases.
  8. Rectify the grammatical and typo errors. Several unnecessary capitalizations throughout the manuscript.
  9. Line 455, page 13, “The use of information technologies, artificial intelligence and analysis through Business Intelligence tools, as has been detailed in this article….”. There is no information about artificial intelligence in this manuscript. Thus, I have to disagree with this claim.

Reviewer 2 Report

Overall an interesting topic, but requires some clarifications on rationale and the results can be presented in a more clear manner. 

Title- the title is not clear, could be more descriptive, and grammar is poor

-Overall the English needs proofreading and editing (i.e. sentences like “The surveys were carried out by trained people, for the case of the questionnaire at hand with specific training in it” [page3, line122] are odd)

Abstract

-One intro sentence would be good

-What do “Laboratory” and “Gynecology” refer to, abstract is unclear

-“The most prevalent assessment was 4 out of 5 points in all the items of the Likert scales of the questionnaire,”—this is not a useful, just include the meaning

-“application of measures based on the use of new technologies” what new technologies?

-The abstract is lacking key information

Intro

Page 2, line 50 “supooses” typo

-The first three paragraphs of the intro are very general and vague—what tools, what new technologies, some clear example would be helpful—or just go straight to the use cases in healthcare systems and expand on those. There is additional literature on telehealth and barriers and issues with it that should be included here

-ICT is being used as a very broad term here but it’s not clear what actual technologies are being studied—is it patient EHR portals, is it telehealth applications, need more details in the intro

-“We carried out an analysis of the 99 response times involved in the study, how they have been affected by the COVID-19 pandemic and how we could shorten the duration of a process, eliminating waiting times by making use of digital media.”—this belongs in the methods or results

Materials and Methods

-What is the “Laboratory” consultation? Needs explanation for those not familiar with the Spanish system

-Page 3, line 115 “TIC” should this be “ICT”?

-There is no statement about ethical board review; did patients have to provide consent?

Statistical Analysis

Page 4, line 167- “Qualitative variables were expressed as a frequency distribution with their 95% confidence intervals”—this does not make sense, how can “qualitative” variables be analyzed? I suspect the authors are using “qualitative” incorrectly, this usually refers to unstructured data (generally interview or focus group data)

Results

Page 5, line 209 “The questions whose average score is greater than 4, considering that the assessment is made on a Likert-type scale from 1 to 5, have been mainly five: P4, P63, P7, P9 and P10.” Sentences like this are not useful, need to put this into context of what was the question and what are most patients responding (i.e. what is 4 on the Likert scale)

-Presenting means for a 5-point Likert scale is not as useful as just presenting the % of respondents for each option—redo Table 2

-Why were the ages grouped as they were? These are large age ranges, what is the utility in grouping this way vs just presenting the mean age

-Table 3—It is not clear why these two groups (gynecology and laboratory) were compared in the first place? These are two completely different indications so it makes sense that the patient populations are different. This rationale should be in the intro or methods

-A Wilcoxon rank sum test may be more appropriate here than the Mann Whitney test.

-There is no P56 in Table 1 but there is in Tables 2 & 3?

-Table 4—why split ages in this way?

Discussion

-Page 8, line 261 “Two types of tests are differentiated in the analysis, one aimed at healthy patients, 261 whose test is for follow-up and control, as is the case of patients who go to the laboratory 262 (EPI pregnancy, childbirth, and puerperium) and the other case is for patients with some 263 risk factor, which are those who attend the Gynecology consultation (PAI cervical cancer).” The conditions in the laboratory group cannot be considered “healthy patients” with no other assessments of comorbities. Also if the intent was to compare views between “healthy patients” and those with active health issues this should have been stated up front in the intro

-Page 10, line 335 “El porcentaje de profesionales sanitarios que ha recomendado páginas 335 web es de sólo del 6,4% y del 7,5% en el caso de las Apps” needs translation

Round 2

Reviewer 1 Report

Authors have addressed all my concerns and paper can be accepted in current form.

Reviewer 2 Report

The manuscript has been improved after revision, overall language still needs editing, but it is minor.